# Association of lymphopenia and RDW elevation with risk of mortality in acute aortic dissection

**Dan Yu** [1,2,3], **Peng Chen**[1], **Xueyan Zhang**[4], **Hongjie Wang**[1,2], **Menaka Dhuromsingh**[1,2], **Jinxiu Wu**[5], **Bingyu Qin**[4]*, **Suping Guo**[3,6]*, **Baoquan Zhang**[5]*, **Chunwen Li**[7]*, **Hesong Zeng**[1,2]*

1 Division of Cardiology, Department of Internal Medicine, Tongji Hospital, Tongji Medical College, Huazhong University of Science and Technology, Wuhan, China, 2 Hubei Provincial Engineering Research Center of Vascular Interventional Therapy, Wuhan, China, 3 Department of Cardiac Intensive Care Unit, People's Hospital of Zhengzhou University (Henan Provincial People's Hospital), Zhengzhou, China, 4 Department of Critical Care Medicine, Henan Key Laboratory for Critical Care Medicine, People's Hospital of Zhengzhou University (Henan Provincial People's Hospital), Zhengzhou, China, 5 Department of Critical Care Medicine, The Third Affiliated Hospital of Xinxiang Medical University, Xinxiang, China, 6 Department of Cardiac Intensive Care Unit, Central China Fuwai Hospital of Zhengzhou University (Fuwai Central China Cardiovascular Hospital), Zhengzhou, China, 7 Department of Emergency Medicine, The Second Affiliated Hospital of Chongqing Medical University, Chongqing, China

* nicolasby@126.com (BQ); gsp389@126.com (SG); pzbaoq@163.com (BZ); chunwenli@cqmu.edu.cn (CL); zenghs@tjh.tjmu.edu.cn (HZ)

**Data Availability Statement:** All relevant data are within the paper and its Supporting Information files.

## Abstract

### Objective

The study aimed to investigate whether lymphopenia and red blood cell distribution width (RDW) elevation are associated with an increased risk of mortality in acute aortic dissection (AAD).

### Methods

This multicenter retrospective cohort study enrolled patients diagnosed with AAD by aortic computed tomographic angiography (CTA) from 2010 to 2021 in five teaching hospitals in central-western China. Cox proportional hazards regression and Kaplan-Meier curves were used in univariable and multivariable models. Clinical outcomes were defined as all-cause in-hospital mortality, while associations were evaluated between lymphopenia, accompanied by an elevated RDW, and risk of mortality.

### Results

Of 1903 participants, the median age was 53 (interquartile range [IQR], 46–62) years, and females accounted for 21.9%. Adjusted increased risk of mortality was linearly related to the decreasing lymphocyte percentage (*P*-non-linearity = 0.942) and increasing RDW (*P*-non-linearity = 0.612), and per standard deviation (SD) of increment lymphocyte percentage and RDW was associated with the 26% (0.74, 0.64–0.84) decrement and 5% (1.05, 0.95–1.15) increment in hazard ratios (HRs) and 95% confidence intervals (CIs) of mortality, respectively. Importantly, lymphopenia and elevation of RDW exhibited a significant interaction with increasing the risk of AAD mortality (*P*-value for interaction = 0.037).

**Funding:** This research was funded by grant CSCF2020B03 from the Chinese Society of Cardiology Foun-dation and the National Natural Science Foundation of China (8187021109, 8207021929, 82100510). The funders had no role in study design, data collection and analysis, decision to publish, or preparation of the manuscript.

**Competing interests:** The authors have declared that no competing interests exist.

## Conclusions

Lymphopenia accompanied by the elevation of RDW, which may reflect the immune dysregulation of AAD patients, is associated with an increased risk of mortality. Assessment of immunological biomarkers derived from routine tests may provide novel perspectives for identifying the risk of mortality.

## Introduction

Acute aortic dissection (AAD) is a potentially lethal condition with high morbidity and mortality [1, 2]. Despite remarkable progress in diagnostic and therapeutic techniques, the global burden of AAD remains high [3]. Assessing clinical indicators associated with the risk of mortality is greatly important for patient management, especially when the indicators can be targeted.

Various indicators have been associated with the outcome of AAD, including but not limited to clinical signs, anatomy, hemodynamics, and a series of biomarkers, such as D-dimer and fibrin degradation products [4]. However, an in-depth investigation of the indicators based on the underlying pathogenesis and mechanisms will be more valuable for improving disease outcomes than such studies. AAD mainly occurs either spontaneously (sporadic) or in association with a genetic condition and trauma. The main mechanism that contributes to sporadic AAD is inflammation [3, 5–8]. Theoretically, inflammation involving the innate and adaptive immune systems is a normal response to injury [9, 10]. However, immune dysregulation manifested as a disordered immune response finally results in widespread inflammation and multiorgan damage [11]. The prognostic implications of immune dysregulation for cardiovascular disease have been extensively studied and well elucidated [12–15], although little attention has been paid to the association between dysregulation of immunologic function and AAD mortality.

It is generally believed that lymphopenia is one of the typical phenotypes of immune dysregulation and is among the strongest risk factors for outcomes in cardiovascular and noncardiovascular disorders [16–21]. However, the relationship between lymphopenia and AAD mortality has rarely been studied. Furthermore, immune pathways often influence multiple variables, some of which are measured routinely on admission and are substantially neglected by clinicians. For instance, evidence has indicated that an elevated red blood cell distribution width (RDW) is associated with cardiovascular and noncardiovascular disease and death [22, 23]. However, to the best of our current knowledge, the extent to which lymphopenia is associated with the risk of mortality in AAD and whether RDW is an additive risk factor beyond lymphopenia have not yet been studied.

In the present study, we characterized the associations among lymphopenia, elevated RDW, and risk of mortality in AAD. We aimed to test the hypothesis that lymphopenia of AAD patients enhances the risk of mortality, and those with concomitantly abnormal elevation of RDW have further reduced survival.

## Materials and methods

### Study population and data collection

We designed a hospital-based multicenter retrospective cohort study by extracting data from electronic medical records (EMRs) of patients admitted to Tongji Hospital Tongji Medical

College of Huazhong University of Science and Technology, People's Hospital of Zhengzhou University, Central China Fuwai Hospital of Zhengzhou University, the Third Affiliated Hospital of Xinxiang Medical University, and the Second Affiliated Hospital of Chongqing Medical University, respectively, which are five teaching hospitals distributed in central-western China. This study followed the Strengthening the Reporting of Observational Studies in Epidemiology (STROBE) reporting guideline for cohort studies.

According to the actual situation of EMRs in different hospitals, we enrolled all available patients who underwent aortic computed tomographic angiography (CTA) between August 2010 and October 2021. Participants who were diagnosed with intramural hematoma (IMH) and penetrating atherosclerotic ulcer (PAU) were excluded, along with participants with a diagnosis of aortic aneurysms, participants with diagnoses including postoperative review of aortic dissection, and participants without a diagnosis of aortic diseases. After applying the first two-round study exclusion criteria, enrolled patients were screened out for further extraction of study variables. The variables belonging to the following categories were extracted separately over the admission period: demographics, initial vital signs, medical history, routine blood tests, coagulation function, biochemical tests, CTA-based anatomical classification, and the status of all-cause in-hospital death. Furthermore, patients aged < 18 years, pregnant aortic dissection patients, non-Han patients, patients with incomplete routine blood tests, patients with disease onset ≥ 14 days, and those with vague records of onset time were further excluded after the collection of variables.

The study was conducted between January 2022 and August 2022, after approval from the Research Ethics Commissions of Tongji Hospital Tongji Medical College of Huazhong University of Science and Technology (TJ- IRB20211102), People's Hospital of Zhengzhou University (2021–190), Central China Fuwai Hospital of Zhengzhou University (2021–38), the Third Affiliated Hospital of Xinxiang Medical University (K2021-039-01), and the Second Affiliated Hospital of Chongqing Medical University (2022–15), respectively, with waived informed consent by the Ethics Commissions mentioned above. The authors had access to information that could identify individual participants during or after the data collection.

## Study variables

The primary outcome of this study was all-cause in-hospital mortality associated with study baseline variables. Immune dysregulation participants were defined as AAD patients characterized by lymphopenia and elevated RDW. The primary exposure was the percentage of lymphocytes, and the secondary exposure was the RDW level. The variables of demographics, initial vital signs, medical history, and the status of all-cause in-hospital death were identified based on the EMRs in different hospitals. All analysis results of routine blood tests, coagulation function, and biochemical tests were obtained from the laboratory department of each hospital and all the blood test samplings were completed within 2 h of admission. The anatomical classification was independently judged by a skilled clinician based on aortic CTA, while the DeBakey system [6] and the category of isolated abdominal AAD [24, 25] were used for identification. The etiology of AAD was defined as genetic, traumatic, congenital disorder, vascular inflammation, infectious disease, and sporadic based on whether the patients had a history of Marfan syndrome, trauma, bicuspid aortic valve, Takayasu arteritis and syphilis.

## Statistical analysis

Baseline characteristics of the analytic sample were summarized across all-cause in-hospital death status as continuous variables and were represented as the median (interquartile range [IQR]), and categorical variables were presented as a percentage. Baseline characteristics were

compared using the Chi-squared test for categorical variables and the Mann–Whitney U test for continuous variables.

Multivariable Cox proportional hazard regression models were used to estimate hazard ratios (HRs) and 95% confidence intervals (CIs) for the associations of lymphocyte percentage and RDW level with all-cause in-hospital mortality. Referring to previous researches [2], all models were successively adjusted for age (continuous), sex (female, male), smoking history (yes, no), hypertension history (yes, no), diabetes history (yes, no), aortic valve replacement history (yes, no), anatomical classification (DeBakey I, DeBakey II, DeBakey IIIa, DeBakey IIIb, or isolated abdominal AAD), etiology (genetic, traumatic, congenital disorder, vascular inflammation, infectious disease, or sporadic), aorta diameter ($\geq 5.5$ cm, $< 5.5$ cm), onset time ($< 24$ h, 1–7 d, 8–14 d), and hospital centers (Tongji Hospital, People's Hospital of Zhengzhou University, Central China Fuwai Hospital of Zhengzhou University, the Third Affiliated Hospital of Xinxiang Medical University, or the Second Affiliated Hospital of Chongqing Medical University). The dose-response curves presenting the hazard of lymphocyte percentage and RDW level were fitted by using the restricted cubic spline model with four knots (rms, hmisc, lattice, and survival packages in R software). Kaplan-Meier methods were performed for survival curve plotting. We further accessed the interaction between lymphopenia and RDW and their synergistic effects on the risk of AAD mortality, and the statistical significance of the interaction was examined by the joint test.

Several secondary analyses were conducted to examine the robustness of our results. Stratified analyses were performed across ages, sexes, smoking history, hypertension history, diabetes history, aortic valve replacement history, anatomical classifications, etiologies, aorta diameter, onset time and hospital centers. We calculated the *p*-value for interaction to examine the consistency of patterns in the main results. Considering the influence of acute kidney injury, procedure of operation, transfusions, stroke or coma, and limb ischemia on AAD mortality, we analyzed the outcomes in prespecified subgroups, divided according to the option of the above-mentioned parameters. Given unavailable records of hyperlipidemia history, we further adjusted for random lipid levels on admission. Considering the potential effects on Cox proportional hazard model fit, propensity score matching (PSM) was performed to adjust for differences in baseline characteristics between in-hospital alive and in-hospital dead groups. Cox proportional hazard regression models were re-fitted in the matched population to test the stability of our results. Missing values of covariates were treated as dummy variables. SAS version 9.4 (SAS Institute, USA) and R software (the R Foundation, http://www.r-project.org, version 4.0.2) were utilized for analyses and plotting with a two-sided significance threshold of $P < 0.05$.

## Results

### Study participants

From all five teaching hospital centers, we identified 35,260 patients who underwent aortic CTA examination between August 2010 and October 2021. In the first round of study participant exclusion, 5214 patients were collected as those diagnosed with first-onset AAD from all five hospitals. Moreover, 2242 AAD inpatients with available EMRs were summarized after the second-round exclusion. In total, 1903 patients were confirmed eligible for statistical analysis at the third-round exclusion. A flowchart of the study, including the details of participants' screening, is shown in **Fig 1**.

### Baseline characteristics

A total of 1903 patients were included in the present analysis in which the median age was 53 (IQR 46–62) years, and females accounted for 21.9%. The baseline characteristics of the

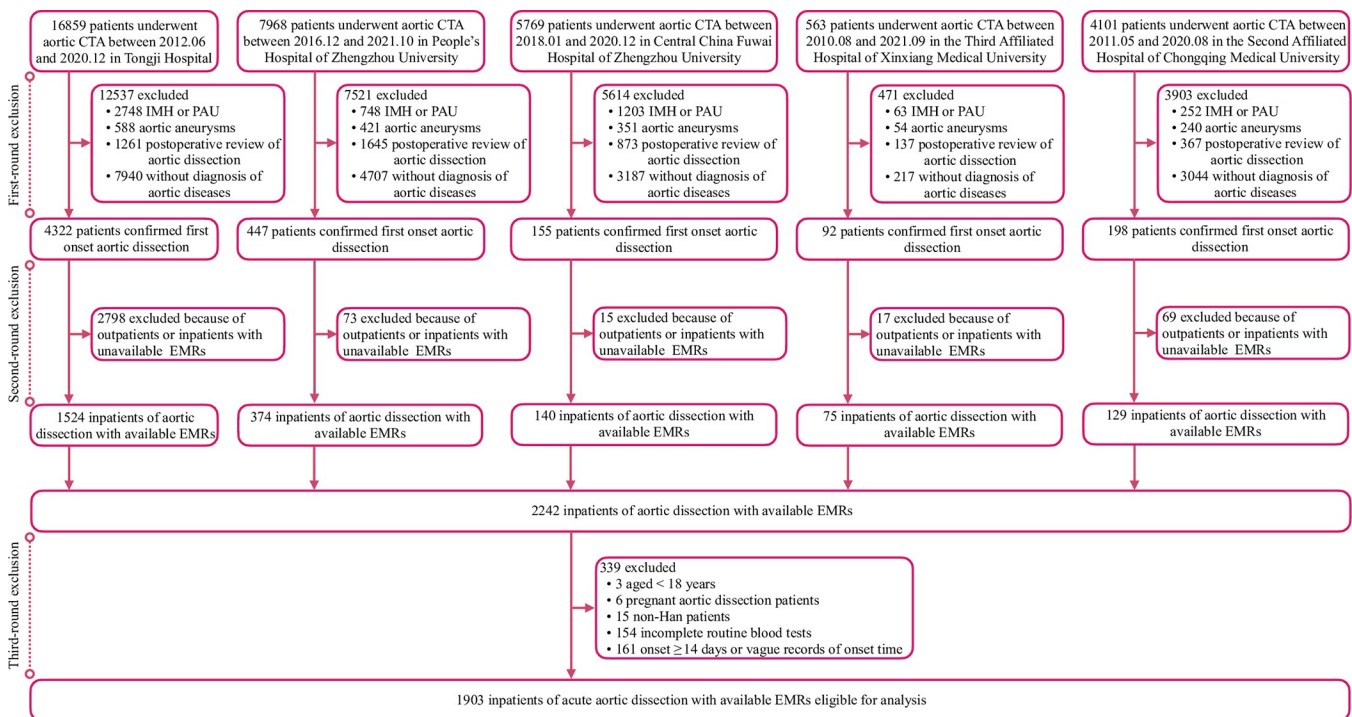

**Fig 1. Flowchart of the study.** CTA, computed tomographic angiography; IMH, intramural hematoma; PAU, aortic pseudoaneurysm; EMRs, electronic medical records.

participants are provided in **Table 1** and **S1 Table**. Among these patients, there were 1430 (75.1%) in-hospital alive and 473 (24.9%) in-hospital dead patients. Generally, those in-hospital dead participants were more likely to be classified as DeBakey I, be a smoker, undergo a surgical operation or had no procedure of operation, onset AAD within 24 h, and had acute kidney injury, stroke or coma, and had limb ischemia. However, there were no significant differences between surviving and deceased patients in age, sex, history of hypertension, diabetes and aortic valve replacement, etiology and transfusions.

### Association of lymphopenia and elevation of RDW with risk of AAD all-cause in-hospital mortality

In Cox regression analyses, lymphocyte percentage and RDW were firstly applied as continuous variables to fitted smoothing splines to present the dose-response relationship between these two variables and the risk of mortality. The adjusted risk of all-cause in-hospital mortality was positively associated with decreasing lymphocyte percentage and increasing RDW (**Fig 2**). The test of non-linearity all-cause in-hospital mortality was not statistically significant for lymphocyte percentage ($P$-non-linearity = 0.942) and RDW ($P$-non-linearity = 0.612). We converted lymphocyte percentage and RDW into categorical variables based on quintiles of the distribution. The unadjusted Kaplan–Meier survival curve demonstrated a significant variance among the groups of patients with the different quintiles categories of lymphocyte percentage and RDW level ($P < 0.0001$, log-rank test) (**Fig 3**).

Stepwise Cox regression analyses were conducted to build adjusted models while adequately considering possible confounders to further explore the risk of AAD all-cause in-hospital mortality according to lymphocyte percentage and RDW level categories. Of note, these

**Table 1. Patient baseline characteristics.**

| Variables | Total patients | In-hospital alive | In-hospital dead | *P*-value |
|---|---|---|---|---|
| N | 1903 | 1430 | 473 | |
| Female, n (%) | 416 (21.9%) | 307 (21.5%) | 109 (23.0%) | 0.472 |
| Age, mean median (IQR), years | 53 (46–62) | 53 (45–62) | 54 (47–62) | 0.056 |
| Anatomical classification | | | | |
| DeBakey I, n (%) | 1021 (53.7%) | 638 (44.6%) | 383 (81.0%) | <0.001 |
| DeBakey II, n (%) | 125 (6.6%) | 100 (7.0%) | 25 (5.3%) | |
| DeBakey IIIa, n (%) | 59 (3.1%) | 57 (4.0%) | 2 (0.4%) | |
| DeBakey IIIb, n (%) | 606 (31.8%) | 549 (38.4%) | 57 (12.1%) | |
| Isolated abdominal AAD, n (%) | 92 (4.8%) | 86 (6.0%) | 6 (1.3%) | |
| Etiology | | | | |
| Genetic (MFS), n (%) | 21 (1.1%) | 14 (1.0%) | 7 (1.5%) | 0.366 |
| Traumatic, n (%) | 23 (1.2%) | 20 (1.4%) | 3 (0.6%) | 0.187 |
| Congenital disorder (BAV), n (%) | 3 (0.2%) | 2 (0.1%) | 1 (0.2%) | 0.576 |
| Vascular inflammation (Takayasu arteritis), n (%) | 3 (0.2%) | 3 (0.2%) | 0 (0.0%) | >0.99 |
| Infectious disease (Syphilis), n (%) | 17 (0.9%) | 14 (1.0%) | 3 (0.6%) | 0.49 |
| Sporadic, n (%) | 1837 (96.5%) | 1378 (96.4%) | 459 (97.0%) | 0.486 |
| History | | | | |
| Smoking, n (%) | 653 (34.3%) | 509 (35.6%) | 144 (30.4%) | 0.041 |
| Hypertension, n (%) | 1172 (61.6%) | 897 (62.7%) | 275 (58.1%) | 0.075 |
| Diabetes, n (%) | 54 (2.8%) | 44 (3.1%) | 10 (2.1%) | 0.274 |
| Aortic valve replacement, n (%) | 15 (0.8%) | 11 (0.8%) | 4 (0.8%) | 0.871 |
| Procedure of operation | | | | |
| None, n (%) | 573 (30.1%) | 316 (22.1%) | 257 (54.3%) | <0.001 |
| Endovascular management, n (%) | 648 (34.1%) | 616 (43.1%) | 32 (6.8%) | |
| Surgical operation, n (%) | 469 (24.6%) | 316 (22.1%) | 153 (32.3%) | |
| Surgical operation and endovascular management, n (%) | 213 (11.2%) | 182 (12.7%) | 31 (6.6%) | |
| Onset time | | | | |
| < 24h, n (%) | 1118 (58.7%) | 802 (56.1%) | 316 (66.8%) | <0.001 |
| 1-7d, n (%) | 692 (36.4%) | 551 (38.5%) | 141 (29.8%) | |
| 8-14d, n (%) | 93 (4.9%) | 77 (5.4%) | 16 (3.4%) | |
| Aorta diameter | | | | |
| ≥ 5.5 cm, n (%) | 1720 (96.6%) | 1298 (97.2%) | 422 (94.8%) | 0.015 |
| < 5.5 cm, n (%) | 60 (3.4%) | 37 (2.8%) | 23 (5.2%) | |
| Acute kidney injury, n (%) | 275 (14.5%) | 147 (10.3%) | 128 (27.1%) | <0.001 |
| Stroke or coma, n (%) | 119 (6.3%) | 53 (3.7%) | 66 (14.0%) | <0.001 |
| Transfusions, n (%) | 723 (38.0%) | 537 (37.6%) | 186 (39.3%) | 0.492 |
| Limb ischemia, n (%) | 163 (8.6%) | 96 (6.7%) | 67 (14.2%) | <0.001 |
| Hospital centers, n (%) | | | | |
| Tongji Hospital | 1345 (70.7%) | 948 (66.3%) | 397 (83.9%) | <0.001 |
| People's Hospital of Zhengzhou University | 253 (13.3%) | 212 (14.8%) | 41 (8.7%) | |
| Central China Fuwai Hospital of Zhengzhou University | 115 (6.0%) | 99 (6.9%) | 16 (3.4%) | |
| Third Affiliated Hospital of Xinxiang Medical University | 67 (3.5%) | 56 (3.9%) | 11 (2.3%) | |
| Second Affiliated Hospital of Chongqing Medical University | 123 (6.5%) | 115 (8.0%) | 8 (1.7%) | |
| Lymphocyte percentage, median (IQR), % | 8.4 (5.5–13.4) | 8.9 (5.9–14.5) | 7.0 (4.9–10.3) | <0.001 |
| RDW-SD, median (IQR), fL | 43.4 (41.1–46.0) | 43.1 (41.0–45.7) | 44.0 (41.8–46.6) | <0.001 |
| Total cholesterol, median (IQR), mmol/L | 3.9 (3.4–4.5) | 3.9 (3.4–4.5) | 3.8 (3.2–4.3) | <0.001 |

(*Continued*)

**Table 1.** (Continued)

| Variables | Total patients | In-hospital alive | In-hospital dead | *P*-value |
|---|---|---|---|---|
| LDL-C, median (IQR), mmol/L | 2.3 (1.7–2.8) | 2.3 (1.7–2.8) | 2.2 (1.7–2.7) | 0.385 |

Continuous variables are represented as median (IQR) and categorical variables as numbers (%).

IQR, interquartile range; AAD

acute aortic dissection; MFS

Marfan syndrome; BAV

Bicuspid aortic valve; RDW-SD, red cell volume distribution width-SD

LDL-C, low density lipoprotein.

associations remained robust after stepwise adjustment for confounders (Table 2). In the crude model, the HRs and 95% CIs from the lowest to the highest lymphocyte percentage categories ($\leq$ 5.0, 5.1–7.1, 7.2–9.7, 9.8–14.8, and $\geq$ 14.9%) were 1.00 (reference), 0.86 (0.66, 1.10), 0.72 (0.55, 0.93), 0.60 (0.46, 0.79), and 0.32 (0.23, 0.44), respectively, for all-cause in-hospital mortality; while the HRs and 95% CIs for the same categories were 1.00 (reference), 0.89 (0.69, 1.15), 0.79 (0.61, 1.03), 0.75 (0.57, 0.99), and 0.50 (0.35, 0.72), respectively, in the fully adjusted model. Additionally, the HRs and 95% CIs from the lowest to the highest RDW categories ($\leq$ 40.7, 40.8–42.4, 42.5–44.0, 44.1–46.7, and $\geq$ 46.8 fL) were 1.00 (reference), 1.33 (0.96, 1.83), 1.37 (1.00, 1.88), 1.96 (1.45, 2.64), and 1.89 (1.40, 2.56), respectively, for all-cause in-hospital mortality in the crude model; while the HRs and 95% CIs for the same categories were 1.00 (reference), 1.16 (0.84, 1.61), 1.17 (0.84, 1.62), 1.51 (1.11, 2.06), and 1.45 (1.06, 1.98), respectively, in the fully adjusted model. When the lymphocyte percentage and RDW were considered a continuous variable, per standard deviation (SD) of increment lymphocyte percentage and RDW was associated with the 26% (0.74, 0.64–0.84) decrement and 5% (1.05, 0.95–1.15) increment in HRs and 95% CIs of AAD all-cause in-hospital mortality, respectively.

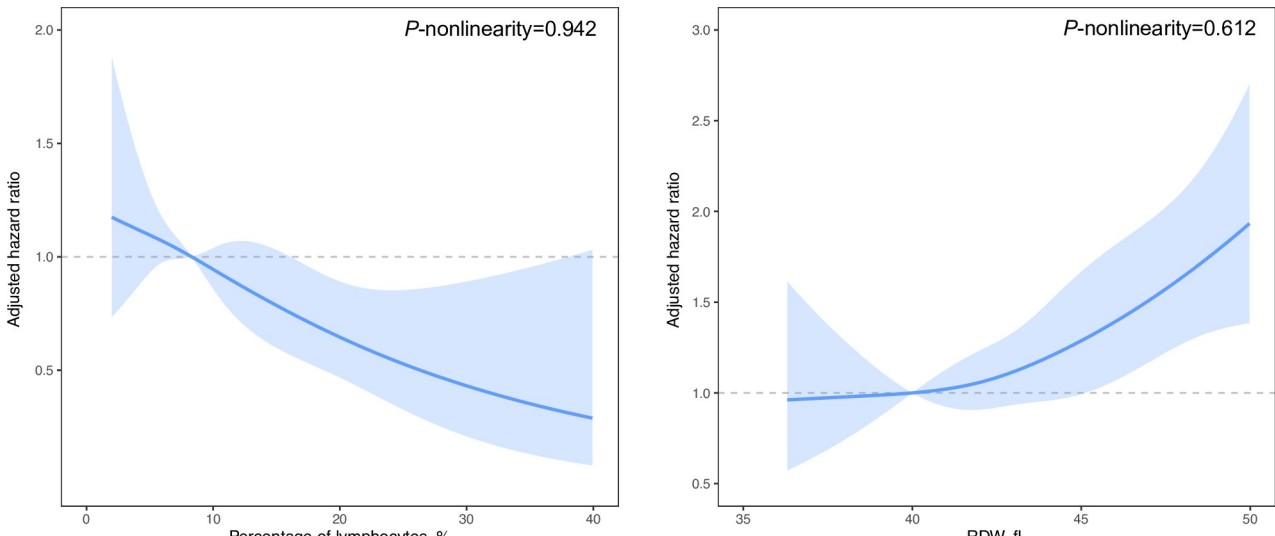

**Fig 2. Dose-response curves for lymphocyte percentage and RDW level with risk of AAD in-hospital mortality.** Hazard ratios (blue lines) and 95% confidence intervals (light blue shade) were adjusted for age, sex, smoking history, hypertension history, diabetes history, aortic valve replacement history, anatomical classification, etiology, aorta diameter, onset time and hospital centers. AAD, acute aortic dissection; RDW, red blood cell distribution width.

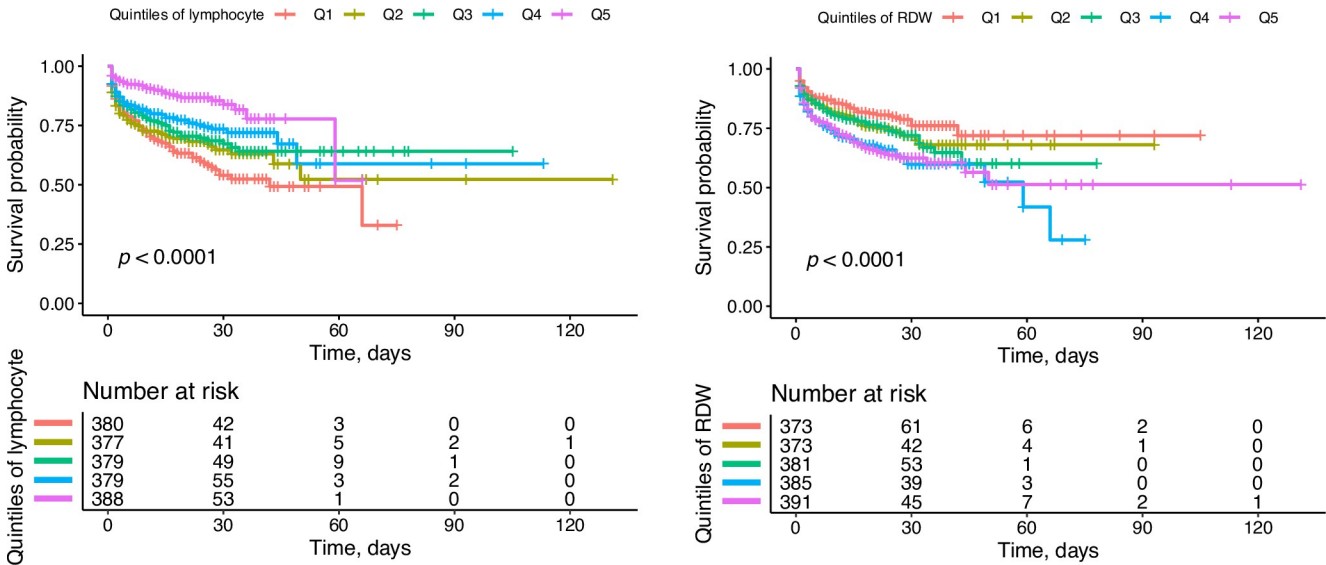

**Fig 3. Kaplan–Meier survival curve of AAD in-hospital mortality showing that outcomes significantly varied among the groups of patients with the different quintiles categories of lymphocyte percentage and RDW level ($P < 0.0001$ log-rank tests).** AAD, acute aortic dissection; RDW, red blood cell distribution width.

## Risk of all-cause in-hospital mortality according to lymphopenia accompanied by elevation of RDW

Apart from identifying the association of lymphopenia and elevation of RDW individually with the risk of AAD all-cause in-hospital mortality, we further explored the outcome of mortality when lymphopenia was accompanied by elevated RDW. Lymphocyte percentage and RDW were again converted from continuous variables into categorical variables according to observed tertiles of the distribution to investigate the interaction among different categories of lymphopenia and elevation of RDW on influencing the risk of AAD mortality. When

**Table 2. Associations of lymphocyte percentage and RDW level with risk of in-hospital mortality.**

| | Quintiles of the exposure | | | | | P trend | Per SD increment |
|---|---|---|---|---|---|---|---|
| | Q1 | Q2 | Q3 | Q4 | Q5 | | |
| Lymphocyte percentage, % | ≤ 5.0 | 5.1–7.1 | 7.2–9.7 | 9.8–14.8 | ≥ 14.9 | | |
| Deaths/N | 133/380 | 111/377 | 99/379 | 83/379 | 47/388 | | |
| Crude model | 1 (reference) | 0.86 (0.66, 1.10) | 0.72 (0.55, 0.93) | 0.60 (0.46, 0.79) | 0.32 (0.23, 0.44) | <0.0001 | 0.62 (0.54, 0.71) |
| Model 1 | 1 (reference) | 0.85 (0.66, 1.10) | 0.73 (0.57, 0.95) | 0.60 (0.46, 0.79) | 0.31 (0.22, 0.43) | <0.0001 | 0.61 (0.54, 0.70) |
| Model 2 | 1 (reference) | 0.89 (0.69, 1.15) | 0.79 (0.61, 1.03) | 0.75 (0.57, 0.99) | 0.50 (0.35, 0.72) | 0.0002 | 0.74 (0.64, 0.84) |
| RDW,fL | ≤ 40.7 | 40.8–42.4 | 42.5–44.0 | 44.1–46.7 | ≥ 46.8 | | |
| Deaths/N | 67/373 | 84/373 | 88/381 | 119/385 | 115/391 | | |
| Crude model | 1 (reference) | 1.33 (0.96, 1.83) | 1.37 (1.00, 1.88) | 1.96 (1.45, 2.64) | 1.89 (1.40, 2.56) | <0.0001 | 1.12 (1.03, 1.21) |
| Model 1 | 1 (reference) | 1.30 (0.95, 1.80) | 1.31 (0.96, 1.81) | 1.84 (1.36, 2.50) | 1.75 (1.29, 2.38) | <0.0001 | 1.09 (1.00, 1.19) |
| Model 2 | 1 (reference) | 1.16 (0.84, 1.61) | 1.17 (0.84, 1.62) | 1.51 (1.11, 2.06) | 1.45 (1.06, 1.98) | 0.0046 | 1.05 (0.95, 1.15) |

Data was reprsented as numbers and HR (95% CI).

Model 1: adjusted for age (continuous) and sex.

Model 2: adjusted for model 1 plus smoking history, hypertension history, diabetes history, aortic valve replacement history, anatomical classification, etiology, aorta diameter, onset time and hospital centers.

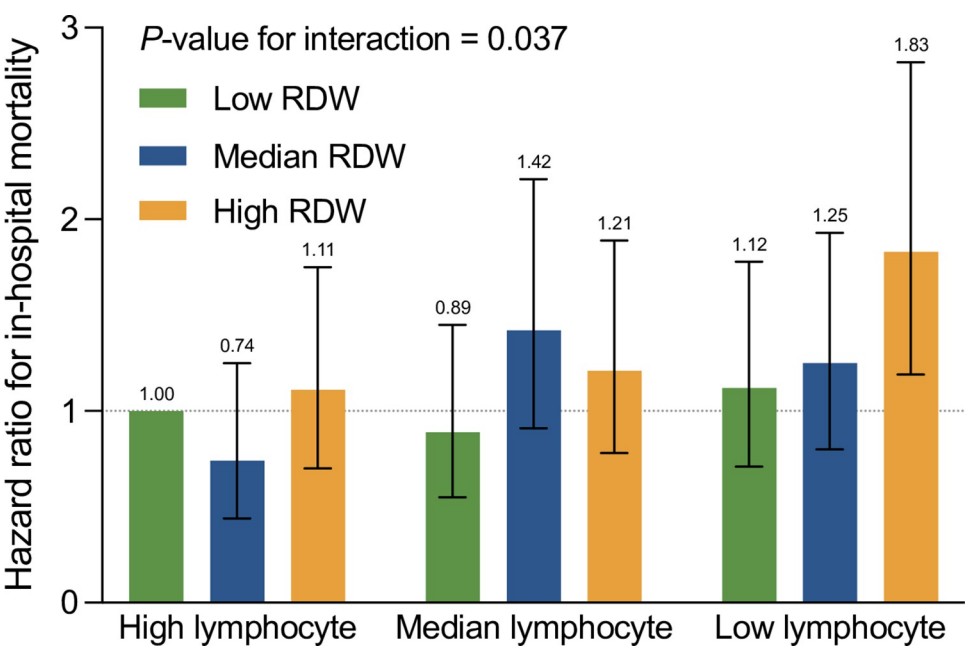

**Fig 4. Risk of in-hospital mortality according to lymphocyte percentage and RDW level.** Hazard ratios were adjusted for age, sex, smoking history, hypertension history, diabetes history, aortic valve replacement history, anatomical classification, etiology, aorta diameter, onset time and hospital centers. RDW, red blood cell distribution width.

lymphocyte percentage and RDW categories were combined, there was an obvious association between decreasing lymphocyte percentage and elevated RDW (**Fig 4** and **Table 3**). Patients with low lymphocyte percentage and high RDW were at a higher risk of all-cause in-hospital mortality versus those with high lymphocyte percentage and low RDW (HR 1.83, 95% CI 1.19–2.82). Collectively, our data demonstrated that lymphopenia combined with the elevation of RDW could serve as a novel candidate for predicting in-hospital AAD mortality.

**Table 3. Risk of In-hospital mortality according to lymphocyte percentage and RDW level.**

| Subgroup | Deaths/N | Hazard Ratio (95% CI) | *P*-value for interaction |
|---|---|---|---|
| High lymphocyte percentage | | | 0.037 |
| Low RDW | 32/204 | 1 (reference) | |
| Median RDW | 26/214 | 0.74 (0.44, 1.25) | |
| High RDW | 47/223 | 1.11 (0.70, 1.75) | |
| Median lymphocyte percentage | | | |
| Low RDW | 37/195 | 0.89 (0.55, 1.45) | |
| Median RDW | 63/213 | 1.42 (0.91, 2.21) | |
| High RDW | 63/220 | 1.21 (0.78, 1.89) | |
| Low lymphocyte percentage | | | |
| Low RDW | 53/207 | 1.12 (0.71, 1.78) | |
| Median RDW | 71/226 | 1.25 (0.80, 1.93) | |
| High RDW | 81/201 | 1.83 (1.19, 2.82) | |

Data was represented as numbers and HR (95% CI) and adjusted for age, sex, smoking history, hypertension history, diabetes history, aortic valve replacement history, anatomical classification, etiology, aorta diameter, onset time and hospital centers.

## Secondary analyses

Several secondary analyses were conducted to examine whether various stratified Cox proportional hazards analyses showed consistent results. As shown in S2 Table, the association of lymphocyte percentage with in-hospital mortality was robust across the strata of age, sex, smoking history, hypertension history, diabetes history, anatomical classification, aorta diameter, onset time, hospital centers, acute kidney injury, procedure of operation, transfusions, stroke or coma, limb ischemia, total cholesterol level, and low-density lipoprotein level. Importantly, patients in the strata undergoing endovascular management showed a stronger association between lymphopenia and in-hospital mortality of AAD patients. When we again performed stratified analyses for RDW according to age, sex, smoking history, hypertension history, diabetes history, aortic valve replacement history, anatomical classification, etiology, aorta diameter, onset time, hospital centers, acute kidney injury, procedure of operation, transfusions, stroke or coma, limb ischemia, total cholesterol level, and low-density lipoprotein level, only diabetes history tended to show an increased risk of all-cause in-hospital mortality, whereas other results were consistent in different variables stratified (S3 Table). Additionally, propensity score–matched population was used to fit Cox proportional hazard regression models for further testing the stability and reliability of the results. The baseline characteristics of the populations generated by PSM appear in S4 Table, the association of lymphopenia and elevated RDW with the risk of in-hospital mortality in AAD was similar with the results in Table 2 (S5 Table).

## Discussion

We sought to explore the association between lymphopenia and elevated RDW of AAD patients and the risk of all-cause in-hospital mortality. Additionally, we aimed to establish the extent to which the associated risk of these two variables is additive. We found that lymphopenia and elevation of RDW were associated with the risk of all-cause in-hospital mortality in AAD patients, and those with these two indicators were at a higher risk of mortality. Taken together, our data suggest that lymphopenia and RDW elevation might reflect immune dysregulation and can be viewed as a multidimensional entity to predict the risk of all-cause in-hospital mortality in AAD patients.

In previous studies, immune and inflammatory mechanisms have been widely considered to mediate cardiovascular diseases and affect their prognosis [26, 27], while the main mechanism of sporadic AAD is inflammation [3, 5–8]. These data prompted us to generate a novel hypothesis that immune dysregulation in AAD patients might have an influence on their outcomes. Immune dysregulation is an overarching term used to characterize an array of autoimmune and inflammatory conditions [28], which manifest as abnormalities of immune molecules and cells. Compared with cytokines and innate immune cells, such as neutrophils and monocyte-macrophages, lymphocytes serving as adaptive immune cells have been more easily neglected by clinicians in their routine practice. In our study, we demonstrated the significant association between lymphopenia and the risk of all-cause in-hospital mortality in AAD patients. The hypothesis that immune dysregulation of AAD patients is related to their prognosis was corroborated and presented by lymphopenia, which has been underappreciated in routine clinical work. Because mortality in the AAD population is mainly driven by noninfectious causes, our study supports the notion that immune status is indeed associated with the outcome of cardiovascular disease. Lymphopenia reflect adverse inflammatory, and the increase of inflammatory mediators such as tumor necrosis factor and interleukin 1β may reduce levels of circulating T cells [29]. Moreover, AAD patients experience stress, resulting in excess levels of cortisol and catecholamine, and abnormal hormone levels also cause

lymphopenia [30, 31]. As to why the immune dysregulation might cause lymphopenia, different mechanisms have been proposed, showing that lymphopenia might be caused by the redistribution of T cells and increased susceptibility of T cells to apoptosis [32–34], while more studies are needed to substantiate these viewpoints.

Besides the abnormality of immune cells, immune dysregulation also leads to impaired erythropoiesis via adverse inflammation [35–37]. RDW characterizes the heterogeneity of circulating red blood cells (RBCs) and has been used to differentiate the causes of anemia [38], while it is also utilized to predict the outcomes of cardiovascular and noncardiovascular diseases [37–40]. Furthermore, previous studies have shown that RDW is associated with the levels of interleukin 6 in heart failure and tumor necrosis factor in coronary artery lesions of Kawasaki disease [41]. Given the above-presented description, RDW manifests as another indicator of immune dysregulation and might predict the prognosis of various disorders. Previous multiple studies have identified strong associations between RDW and clinical prognosis in various populations with cardiovascular and noncardiovascular diseases [42–44]. As for the mechanisms, inflammation might lead to altered iron homeostasis and erythropoietin resistance, ultimately causing the elevation of RDW [45, 46]. In this study, the elevation of RDW was identified to be significantly associated with the risk of all-cause in-hospital mortality in AAD patients, which further confirmed the hypothesis of the association between immune dysregulation and the prognosis of AAD patients.

To our knowledge, a small number of related studies have been previously performed, demonstrating similar conclusions to ours. Wei Luo et al. [34] have reported that lymphopenia correlated with poor outcomes in type A aortic dissection patients undergoing surgery. The researchers enrolled a total of 335 patients from two hospitals in one southern Chinese city and found that pre-operative lymphopenia, particularly CD4+ T lymphopenia, correlated with poor prognosis via apoptosis. Additionally, Cheng Jiang et al. [47] have stated that increased RDW was associated with poor outcomes in type B aortic dissection patients undergoing endovascular aortic repair. The investigators included a total of 678 patients from three hospitals in Guangdong province in south China and found that an RDW of > 13.5% on admission was independently associated with increased long-term mortality. Compared to the present study, both of the above-mentioned studies circumscribed the characteristic of the included research patients. Only patients with a specific anatomical classification and undergoing a specific treatments were enrolled in the studies. However, study populations chosen with special features might have limited external generalizability of their results. In the current study, a larger research population was enrolled from a broader territory in China. Patients with all types of aortic dissections based on the anatomical classification and treatments were included to investigate the association between the above-described two indicators and prognosis, and we identified the suitability of results for patients with different features by several secondary analyses. Furthermore, we creatively found a significant interaction between lymphopenia and elevation of RDW for jointly predicting the risk of all-cause in-hospital mortality in AAD patients, and there should be an underlying logical link among them. Theoretically, immune dysregulation might result in abnormal changes of immune cells and immune molecules. In addition to the abnormal changes of immune cells, lymphopenia might reflect adverse inflammatory, metabolic, or neuroendocrine stressors [29]. However, abnormal changed immune molecules, also regarded as inflammatory cytokines, together with activated neurohumoral and adrenergic systems, are both involved in the pathophysiology of RDW elevation in cardiovascular and cerebrovascular diseases [48]. Taken together, the combination of lymphopenia and elevation of RDW might represent a more severe immune phenotype to help in identifying the extremely high-risk AAD patients. However, further research is needed to unveil the potential causes and underlying mechanisms.

Similar to the association between lymphopenia and elevated RDW of AAD patients and their prognosis, hemogram parameters are also important in the evaluation of the other cardiovascular diseases. The prognostic value of hematological indices in stable coronary artery disease (CAD) has demonstrated that the mean platelet volume (MPV) and MPV-to-platelet ratio (MPR) might be associated with the degree of collateral development (CCD) in chronic stable CAD. However, the negative association was also found between RDW and inadequate CCD [49]. Further studies have elucidated those similar indicators including MPV-to-lymphocyte ratio and platelet distribution width (PDW) could serve as a marker for CCD in patients with stable angina and non-ST-elevation myocardial infarction (NSTEMI) [50, 51]. Together, our work and the above-mentioned studies illustrated the extensive clinical value of hemogram parameters in assessing cardiovascular diseases, warranting clinical promotion and popularization.

Aortic dissection is literally defined as the results of the separation (dissection) of aortic wall layers and is caused by a tear in the intimal layer of the aorta. Once the structural properties of the aorta are compromised, existing dissections are aggravated by mechanical stress caused by blood flow until death [6]. Most of life-threatening clinical features that have been reported in the international registry of AAD [2], such as hypotension and shock, pericardial effusion and tamponade, and periaortic hematoma and brain injury, are caused by aortic rupture, for which the potential reason is compromised aorta. Immune dysregulation, which has been previously considered as a disordered immune response resulting in widespread inflammation and multiorgan damage [11], is also involved in the incidence and development of compromised aorta. As to the underlying mechanisms, several immune-related mechanisms have been suggested in previous studies. Firstly, an immune infiltrate has been found within the middle and outer tunics of dissected aortic specimens [52]. Secondly, single-cell transcriptome analysis has revealed dynamic cell populations in control and aneurysmal human aortic tissue in previous study. The tissues of thoracic aortic aneurysm which can lead to aortic dissection, rupture, and other life-threatening complications had fewer nonimmune cells and more immune cells, especially T lymphocytes, than control tissues [53]. Thirdly, the recall and activation of macrophages inside the middle tunic have also been observed and considered as key events in the early phases of AAD, and macrophages could release metalloproteinases (MMPs) and pro-inflammatory cytokines which lead to matrix degradation. Finally, the imbalance between the production of MMPs and MMP tissue inhibitors is pivotal in the extracellular matrix degradation underlying aortic wall remodeling and tearing in dissections [52]. However, more researches is still needed to explore the relationship between immune dysregulation and ADD-related mortality in the future.

Several limitations of our study deserve special consideration. First, this was a retrospective observational study. Although the multicenter cohort study represents a large sample in which we were able to minimize bias, we could not exclude residual confounding, especially by unavailable variables. Second, as with all observational studies, the extent to which these associations might be causal in nature could not be assessed. Third, due to the absent data of follow-up data, we could not evaluate the long-term prognosis of AAD patients and analyze the secondary outcomes, such as relapse of aortic dissection and developing coronary artery and cerebrovascular diseases, and other various complications. Fourth, we assessed death by the discharge record of EMRs, and the specific cause of death could not be clarified with some patients; hence, all-cause in-hospital mortality was the only outcome for analysis in this study. Fifth, as only the Chinese Han population was enrolled in the study, ethnic differences might limit the external generalizability of the results. Despite these limitations, we believe that this study provides beneficial support for clinicians in identifying AAD patients at increased risk of death.

## Conclusions

In this multicenter retrospective cohort study, we investigated the association among lymphopenia and elevation of RDW in AAD patients and the risk of all-cause in-hospital mortality, while Lymphopenia and RDW elevation might suggest immune dysregulation and be used to identify and quantify immunologic abnormalities in the AAD population. The presence of lymphopenia and elevation of RDW was associated with significantly increased mortality, and the risk was further heightened when these two indicators were combined. Assessment of immune status derived from routine tests, which have been severely neglected by clinicians in daily clinical practice, might provide novel perspectives for identifying the risk of mortality. More studies are required to confirm whether the earlier immune intervention might reduce the mortality of AAD by alleviating lymphopenia and elevated RDW.

## Supporting information

**S1 Checklist. STROBE statement—checklist of items that should be included in reports of *cohort studies.***
(DOCX)

**S1 Table. Patient baseline characteristics (continued).**
(DOCX)

**S2 Table. Stratified analyses of the associations of lymphocyte percentage with in-hospital mortality.**
(DOCX)

**S3 Table. Stratified analyses of the associations of RDW with In-hospital mortality.**
(DOCX)

**S4 Table. Patient baseline characteristics (propensity score–matched population).**
(DOCX)

**S5 Table. Associations of lymphocyte percentage and RDW level with risk of in-hospital mortality (propensity score–matched population).**
(DOCX)

## Acknowledgments

The authors thank Dr. Liangkai Chen for his assistance with the statistical analyses.

## Author Contributions

**Conceptualization:** Dan Yu, Hongjie Wang, Bingyu Qin, Suping Guo, Baoquan Zhang, Chunwen Li, Hesong Zeng.

**Data curation:** Xueyan Zhang, Jinxiu Wu.

**Funding acquisition:** Peng Chen, Hesong Zeng.

**Investigation:** Dan Yu, Peng Chen, Xueyan Zhang, Menaka Dhuromsingh, Jinxiu Wu, Suping Guo, Chunwen Li.

**Methodology:** Peng Chen, Baoquan Zhang.

**Resources:** Bingyu Qin, Baoquan Zhang, Chunwen Li, Hesong Zeng.

**Writing – original draft:** Dan Yu.

**Writing – review & editing:** Hongjie Wang, Menaka Dhuromsingh.

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
