## [Decision Letter · Decision Letter 0]

14 Dec 2022

PONE-D-22-29441Association of Lymphopenia and RDW Elevation with Risk of Mortality in Acute Aortic DissectionPLOS ONE

Dear Dr. Zeng,

Thank you for submitting your manuscript to PLOS ONE. After careful consideration, we feel that it has merit but does not fully meet PLOS ONE’s publication criteria as it currently stands. Therefore, we invite you to submit a revised version of the manuscript that addresses the points raised during the review process.

ACADEMIC EDITOR: Authors are asked to revise their paper in accordance with the reviewers' comments. The paper would be re considered if adequately revised.

We look forward to receiving your revised manuscript.

Kind regards,

Gulali Aktas

Academic Editor

PLOS ONE

Journal Requirements:

"This research was funded by grant CSCF2020B03 from the Chinese Society of Cardiology Foun-dation and the National Natural Science Foundation of China (8187021109, 8207021929, 82100510)."

Additional Editor Comments:

Authors are asked to revise their paper in accordance with the reviewers' comments.

Reviewers' comments:

Reviewer's Responses to Questions

**Comments to the Author**

1. Is the manuscript technically sound, and do the data support the conclusions?

Reviewer #1: Yes

Reviewer #2: Yes

2. Has the statistical analysis been performed appropriately and rigorously? 

Reviewer #1: Yes

Reviewer #2: Yes

3. Have the authors made all data underlying the findings in their manuscript fully available?

Reviewer #1: Yes

Reviewer #2: Yes

4. Is the manuscript presented in an intelligible fashion and written in standard English?

Reviewer #1: Yes

Reviewer #2: No

5. Review Comments to the Author

Reviewer #1: 1. Some of the references used in the article are more than 10 years old, and it would be good if possible to re-edit these references with current literature information.

2. English is used at an intermediate level in the article.

Reviewer #2: 1-The topic of the article is interesting.

2-Abstraction of the text is well prepared.

3- The introduction is sufficiently.

4- The methods are unadequate . The difference in numbers between the groups is very large. It must be balanced.

5-The results are enough.

6- The discussion is very short.it should be confused. Its mechanisms should be discussed.

7- The Tables and the figures are sufficient.

8- Also, -, Discussion must contain comparison of similar studies that found association between The Hemogram Parameters and other cardiovasculer diseases diseases (i.e. " Association of mean platelet volume and red blood cell distribution width with coronary collateral development in stable coronary artery disease.DOI:https://doi.org/10.5114/aic.2018.78329.)

9-Please format references according to the journal style.

Major revision

6. PLOS authors have the option to publish the peer review history of their article (what does this mean?). If published, this will include your full peer review and any attached files.

Reviewer #1: No

Reviewer #2: **Yes: **no

---

## [Author Response · Author response to Decision Letter 0]

26 Jan 2023

Dear Academic Editor Gulali Aktas, 

Please find enclosed a revised version of the manuscript PONE-D-22-29441 (PLOS ONE) with the title “Association of Lymphopenia and RDW Elevation with Risk of Mortality in Acute Aortic Dissection”. 

We were pleased by the reviewers’ comments and their constructive suggestions for the improvement of our manuscript. We included changes and new data based on the reviewers’ comments within the revised version. We feel that the new data generated following the suggestions supports our conclusions and strengthen the manuscript. The changes addressing the points raised by the reviewers were shown as track changes in the marked-up copy. 

The manuscript were also revised to meet PLOS ONE's style requirements and the captions for supporting information files were listed at the end of our manuscript.

This research was funded by grant CSCF2020B03 from the Chinese Society of Cardiology Foun-dation and the National Natural Science Foundation of China (8187021109, 8207021929, 82100510) and the funders had no role in study design, data collection and analysis, decision to publish, or preparation of the manuscript.

Appended is our detailed point-by-point response to the reviewers’ comments. We are looking forward to hearing from you. 

Sincerely, 

Hesong Zeng 

Reviewer 1

We thank the reviewer for her/his valuable and constructive suggestions and comments which we believe significantly improved the quality of the manuscript. We have addressed all the comments point by point as follows: 

Comment 1: 

Some of the references used in the article are more than 10 years old, and it would be good if possible to re-edit these references with current literature information.

Answer 1: 

We thank the reviewer for pinpointing this issue and we apologize for the inadvertence. After carefully checked the references in our manuscript, 6 articles which were published over 10 years ago were re-edited with current literature information. The updated and the original references were listed below.

Original citations:

8. Luo F, Zhou XL, Li JJ, Hui RT. Inflammatory response is associated with aortic dissection. Ageing Res Rev. 2009;8(1):31-5. https://doi.org/10.1016/j.arr.2008.08.001 PMID: 18789403

9. Dinarello CA. Anti-inflammatory Agents: Present and Future. Cell. 2010;140(6):935-50. https://doi.org/10.1016/j.cell.2010.02.043 PMID: 20303881

10. Quintans J. Immunity and inflammation: the cosmic view. Immunol Cell Biol. 1994;72(3):262-6. https://doi.org/10.1038/icb.1994.39 PMID: 8088865

22. Tonelli M, Sacks F, Arnold M, Moye L, Davis B, Pfeffer M, et al. Relation Between Red Blood Cell Distribution Width and Cardiovascular Event Rate in People with Coronary Disease. Circulation. 2008;117(2):163-8. https://doi.org/10.1161/CIRCULATIONAHA.107.727545 PMID: 18172029

27. Hansson GK, Hermansson A. The immune system in atherosclerosis. Nat Immunol. 2011;12(3):204-12. https://doi.org/10.1038/ni.2001 PMID: 21321594

37. Allen LA, Felker GM, Mehra MR, Chiong JR, Dunlap SH, Ghali JK, et al. Validation and potential mechanisms of red cell distribution width as a prognostic marker in heart failure. J Card Fail. 2010;16(3):230-8. https://doi.org/10.1016/j.cardfail.2009.11.003 PMID: 20206898

Changed citations:

8. Shen YH, LeMaire SA, Webb NR, Cassis LA, Daugherty A, Lu HS. Aortic Aneurysms and Dissections Series. Arterioscler Thromb Vasc Biol. 2020;40(3):e37-e46. https://doi.org/10.1161/ATVBAHA.120.313991 PMID: 32101472

9 Sun L, Wang X, Saredy J, Yuan Z, Yang X, Wang H. Innate-adaptive immunity interplay and redox regulation in immune response. Redox Biol. 2020;37:101759. https://doi.org/10.1016/j.redox.2020.101759 PMID: 33086106

10. Chen L, Deng H, Cui H, Fang J, Zuo Z, Deng J, et al. Inflammatory responses and inflammation-associated diseases in organs. Oncotarget. 2018;9(6):7204-18. https://doi.org/10.18632/oncotarget.23208 PMID: 29467962

22. Danese E, Lippi G, Montagnana M. Red blood cell distribution width and cardiovascular diseases. J Thorac Dis. 2015;7(10):E402-11. https://doi.org/10.3978/j.issn.2072-1439.2015.10.04 PMID: 26623117

27. Lawler PR, Bhatt DL, Godoy LC, Luscher TF, Bonow RO, Verma S, et al. Targeting cardiovascular inflammation: next steps in clinical translation. Eur Heart J. 2021;42(1):113-31. https://doi.org/10.1093/eurheartj/ehaa099 PMID: 32176778

40. Talarico M, Manicardi M, Vitolo M, Malavasi VL, Valenti AC, Sgreccia D, et al. Red Cell Distribution Width and Patient Outcome in Cardiovascular Disease: A ''Real-World'' Analysis. J Cardiovasc Dev Dis. 2021;8(10). https://doi.org/10.3390/jcdd8100120 PMID: 34677189

Comment 2:

English is used at an intermediate level in the article.

Answer 2:

We thank the reviewer to bring this issue up and apologize for the insufficient English expression level in our manuscript. We have now worked on both language and readability and have also involved native English speakers for language corrections. We really hope that the flow and language level have been substantially improved.

Reviewer 2

We thank all the comments that are valuable and very helpful for revising and improving our paper, as well as the important guiding significance to our researches. We have studied comments carefully and have made correction which we hope meet with approval. The following are the responses and revisions we have made in response to reviewer’s questions and suggestions on an item-by-item basis.

Comment 1: 

The topic of the article is interesting.

Answer 1:

We highly appreciate the positive evaluation.

Comment 2: 

Abstraction of the text is well prepared.

Answer 2:

We thank the reviewer for her/his positive evaluation.

Comment 3: 

The introduction is sufficiently.

Answer 3:

Thank you for the positive comments and evaluation of our study.

Comment 4: 

The methods are unadequate. The difference in numbers between the groups is very large. It must be balanced.

Answer 4:

We thank the reviewer for the careful review of our manuscript and considerate comments. As mentioned by the reviewer that it is indeed there is some difference between the groups in table 1, while we only aimed to present the real baseline characteristics of enrolled patients. when we focused on exploring the association of lymphopenia and RDW elevation with risk of mortality in acute aortic dissection, stepwise multivariable Cox proportional hazard regression models were performed to investigate the association and all the models were successively adjusted for age (continuous), sex (female, male), smoking history (yes, no), hypertension history (yes, no), diabetes history (yes, no), aortic valve replacement history (yes, no), anatomical classification (DeBakey Ⅰ, DeBakey Ⅱ, DeBakey Ⅲa, DeBakey Ⅲb, or isolated abdominal AAD), etiology (genetic, traumatic, congenital disorder, vascular inflammation, infectious disease, or sporadic), aorta diameter (≥ 5.5 cm, < 5.5 cm), onset time (< 24h, 1-7d, 8-14d), and hospital centers (Tongji Hospital, People’s Hospital of Zhengzhou University, Central China Fuwai Hospital of Zhengzhou University, the Third Affiliated Hospital of Xinxiang Medical University, or the Second Affiliated Hospital of Chongqing Medical University). We even performed stratified analyses across ages, sexes, smoking history, hypertension history, diabetes history, aortic valve replacement history, anatomical classifications, etiologies, aorta diameter, onset time and hospital centers to calculate the p-value for interaction to examine the consistency of patterns in the main results. Of note, these associations remained robust after stepwise adjustment for confounders and stratified analyses. Additionally, we added analysis with the propensity score matching (PSM) method. PSM was performed to adjust for differences in baseline characteristics between in-hospital alive and in-hospital dead groups. Cox proportional hazard regression models were re-fitted in the matched population to test the stability and reliability of our results. Eventually, the association of lymphopenia and elevated RDW with the risk of in-hospital mortality in AAD was similar with the results in Table 2.

Comment 5: 

The results are enough.

Answer 5:

We thank the reviewer for this positive evaluation.

Comment 6: 

The discussion is very short. it should be confused. Its mechanisms should be discussed.

Answer 6:

We appreciate the thoughtful comment of the reviewer. we have revised the manuscript and added the description of mechanisms exploration in the second and third paragraphs of discussion section.

Comment 7: 

The Tables and the figures are sufficient.

Answer 7:

We do thank the reviewer’s positive evaluation.

Comment 8: 

Also, discussion must contain comparison of similar studies that found association between the hemogram parameters and other cardiovascular diseases (i.e., " Association of mean platelet volume and red blood cell distribution width with coronary collateral development in stable coronary artery disease.DOI:https://doi.org/10.5114/aic.2018.78329.)

Answer 8:

We thank the reviewer for pointing out this critical point and giving us the constructive suggestion. We have added a paragraph in discussion to elaborate the association between the hemogram parameters and other cardiovascular diseases and citated 3 studies to support our viewpoint (“Sincer I, Gunes Y, Mansiroglu AK, Cosgun M, Aktas G. Association of mean platelet volume and red blood cell distribution width with coronary collateral development in stable coronary artery disease. Postepy Kardiol Interwencyjnej. 2018;14(3):263-9. https://doi.org/10.5114/aic.2018.78329 PMID: 30302102”, ” Ornek E, Kurtul A. Relationship of mean platelet volume to lymphocyte ratio and coronary collateral circulation in patients with stable angina pectoris. Coron Artery Dis. 2017;28(6):492-7. https://doi.org/10.1097/MCA.0000000000000530 PMID: 28678144”, ” Sincer I, Mansiroglu AK, Aktas G, Gunes Y, Kocak MZ. Association between Hemogram Parameters and Coronary Collateral Development in Subjects with Non-ST-Elevation Myocardial Infarction. Rev Assoc Med Bras (1992). 2020;66(2):160-5. https://doi.org/10.1590/1806-9282.66.2.160 PMID: 32428150”). We have tried our best to illustrate the extensively clinical value of hemogram parameters in assessing different cardiovascular diseases in our revised manuscript.

Comment 9: 

Please format references according to the journal style.

Answer 9:

We thank this reviewer to bring this issue up. After careful inspection of the format of references in our manuscript, we revised all the citations to meet PLOS ONE's style requirements.

---

## [Decision Letter · Decision Letter 1]

1 Mar 2023

Association of Lymphopenia and RDW Elevation with Risk of Mortality in Acute Aortic Dissection

PONE-D-22-29441R1

Dear Dr. Zeng,

We’re pleased to inform you that your manuscript has been judged scientifically suitable for publication and will be formally accepted for publication once it meets all outstanding technical requirements.

Kind regards,

Gulali Aktas

Academic Editor

PLOS ONE

Additional Editor Comments (optional):

Reviewers' comments:

Reviewer's Responses to Questions

**Comments to the Author**

1. If the authors have adequately addressed your comments raised in a previous round of review and you feel that this manuscript is now acceptable for publication, you may indicate that here to bypass the “Comments to the Author” section, enter your conflict of interest statement in the “Confidential to Editor” section, and submit your "Accept" recommendation.

Reviewer #1: All comments have been addressed

Reviewer #2: All comments have been addressed

2. Is the manuscript technically sound, and do the data support the conclusions?

Reviewer #1: Yes

Reviewer #2: Yes

3. Has the statistical analysis been performed appropriately and rigorously? 

Reviewer #1: Yes

Reviewer #2: Yes

4. Have the authors made all data underlying the findings in their manuscript fully available?

Reviewer #1: Yes

Reviewer #2: Yes

5. Is the manuscript presented in an intelligible fashion and written in standard English?

Reviewer #1: Yes

Reviewer #2: Yes

6. Review Comments to the Author

Reviewer #1: The article is very well revised in accordance with the previous suggestions. I recommend publication.

Reviewer #2: Dear Editor

I carefully read the article titled "Association of Lymphopenia and RDW Elevation with Risk of Mortality in Acute Aortic Dissection ". The topic of the article is really interesting. Abstraction of the text is well prepared. The introduction is sufficiently a.The methods are adequate. Therefore, results of the study are enought .The discussion is satisfactory . For this reasons, I recommend publication of the article.

7. PLOS authors have the option to publish the peer review history of their article (what does this mean?). If published, this will include your full peer review and any attached files.

Reviewer #1: **Yes: **Tuba Taslamacioglu Duman

Reviewer #2: No

---

## [Editor Report · Acceptance letter]

6 Mar 2023

PONE-D-22-29441R1 

Association of Lymphopenia and RDW Elevation with Risk of Mortality in Acute Aortic Dissection 

Dear Dr. Zeng:

I'm pleased to inform you that your manuscript has been deemed suitable for publication in PLOS ONE. Congratulations! Your manuscript is now with our production department. 

Kind regards, 

on behalf of

Professor Gulali Aktas 

Academic Editor

PLOS ONE